# No Dark Data Required: Bridging the Gap Between Normal and Low-light Detection via Retinex Decomposition

## Abstract

Conventional low-light object detection approaches typically involve distinct image enhancement modules before the detection process. This can lead to compromised performance due to misaligned objectives and reduced robustness in challenging visual contexts. Many existing methodologies either do not optimize both tasks jointly or overlook significant latent features that are essential for accurate detection. To address this issue, a novel end-to-end framework was proposed that was exclusively trained on normal-light images, eliminating the need for low-light data during the training phase. This approach drew inspiration from the Retinex theory, which separated images into reflectance (representing scene structure) and illumination (indicating lighting conditions). The proposed framework approximates this decomposition within the feature space. The architecture utilizes deep multi-scale feature aggregation along with a reflectance-guided fusion pathway, enabling the adaptive integration of illumination-aware representations through element-wise modulation. Despite being trained on normal-light images, the framework demonstrates effective generalization to low-light and visibility-compromised environments. Comprehensive experiments conducted on both synthetic datasets (Pascal VOC) and real-world benchmarks (ExDark, RTTS) indicate that this method achieves enhanced detection accuracy and robustness, particularly in adverse lighting conditions, and outperforms current state-of-the-art techniques.

## 1 Introduction

Operational effectiveness in autonomous systems relies on accurate detection of key objects like pedestrians and vehicles. Contemporary object detection models have achieved significant advancements under optimal conditions; however, their performance often declines in challenging visual environments characterised by inadequate lighting, heterogeneous illumination, and adverse weather conditions. These scenarios can lead to reduced contrast, increased noise, and substantial information loss, ultimately affecting the reliability of leading detection algorithms. Traditional object detection architectures predominantly follow a sequential pipeline where image enhancement techniques precede the detection phase Huang et al. (2021); Kalwar et al. (2022). This configuration may lead to misalignment between enhancement and detection objectives, potentially compromising performance and adaptability in complex visual scenarios. The enhancement module is typically optimised for visual quality, focusing on noise reduction and contrast enhancement, while the detection module is concerned with object localization and classification. This disparity can result in compounded errors, where enhancements may inadvertently introduce artifacts or obscure essential features needed for semantic analysis. As a consequence, improvements in the enhancement module alone may not adequately address these challenges, indicating a need for a systematic integration of enhancement and detection processes to optimise both tasks simultaneously.

Prior study has endeavoured to address object detection in adverse conditions, revealing several limitations.

**Two-Stage Pipelines: Enhancement Followed by Detection** Early methods primarily relied on sequential 'two-stage' pipelines, utilising models like RetinexNet Wei et al. (2018) or Zero-

DCE Guo et al. (2020) for low-light enhancement, followed by detection networks such as YOLO or Faster R-CNN. While these methods can enhance detection accuracy under challenging lighting conditions, they may be prone to feature mismatches and error propagation. When enhancement processes are optimised independently from detection objectives, the risk of generating artefacts or obscuring critical features necessary for robust semantic analysis increases. Even physics-based enhancement strategies may struggle to adapt to the varying relevance of features required for object recognition across diverse low-light conditions.

**Existing End-to-End Approaches:** In response to these challenges, recent research has shifted toward integrated, end-to-end frameworks that optimise both enhancement and detection simultaneously. Models such as MPE-DETR Xue et al. (2024) and FSDNet Hu et al. (2021) aim to learn shared or decomposed feature representations, aligning low-level reconstruction with high-level recognition tasks. Nonetheless, many of these architectures utilize static fusion methodologies, including direct concatenation or summation of enhanced features with their unmodified counterparts Sindagi et al. (2020); Zhang et al. (2021c).

Static fusion techniques can create bottlenecks, limiting the network's adaptability and expressive capacity when faced with diverse visual challenges. By treating all input features uniformly or employing fixed integration methods, static approaches may inadequately adjust the importance of various features—such as original pixel data, enhanced features, and illumination or reflectance components—based on scene-specific degradation. This lack of flexibility can hinder the network's capacity to leverage the most informative cues under varying conditions, indicating the limitations of a 'one size fits all' strategy and highlighting the necessity for context-aware methodologies.

Additionally, investigations of transformer-based architectures, such as LLFormer Wang et al. (2023) and Retinexformer Cai et al. (2023), have not fully addressed the constraints associated with CNN-based sequential processing and may not effectively separate illumination from reflectance information. Furthermore, existing implementations of Retinex-based decomposition often exhibit limitations in task-specific feature integration within the detection framework, underscoring a need for potential enhancement.

Therefore, this study aimed to propose and verify a Retinex-Feature-Decomposition-based YOLO (RFD-YOLO) model, an Artificial Intelligence (AI) solution designed for robust object recognition across diverse conditions that avoids the need for extensive retraining when training datasets are modified to simulate low-light and low-visibility scenarios. Grounded in Retinex theory, RFD-YOLO offers an end-to-end object detection solution designed to address the limitations of current methodologies.

The academic contribution of this AI model lies in its ability to decompose images into reflectance (as an illumination-invariant object property) and illumination based on Retinex theory. This decomposition enables the model to distinguish objects based on their intrinsic structural characteristics rather than the varying illumination conditions. Well-illuminated images provide rich reflectance information, allowing the model to effectively extract and utilize a robust representation for object recognition tasks.

## 2 RELATED WORKS

### 2.1 OBJECT DETECTION IN LOW-LIGHT IMAGES

Study in low-light image enhancement often focuses on improving visual quality, which can hinder recognition accuracy. Models generally fall into three frameworks: Retinex theory-based models, multi-scale and attention-based fusion frameworks Cui et al. (2022); He et al. (2023), and transformer architectures Wei et al. (2018); Wang et al. (2019).

Retinex theory-based models ur Rahman et al. (2004), such as RetinexNet and KinD++ Zhang et al. (2021b), enhance low-light images while addressing issues like noise and colour distortion. EnlightenGAN Jiang et al. (2021) employs unsupervised GAN techniques, whereas Zero-DCE Guo et al. (2020) introduces a lightweight deep curve estimation method for image enhancement.

Multi-scale and attention-based fusion models improve enhancements through feature representation and attention mechanisms. Transformer architectures, exemplified by LYT-NET Brateanu et al.

(2024), represent a shift towards advanced enhancement strategies. However, a significant challenge remains: many models prioritize visual quality, risking the loss of critical detection features and introducing artifacts.

## 2.2 OBJECT DETECTION IN ADVERSARIAL VISIBILITY CONDITIONS

Low-vision image processing, a crucial aspect for real-world applications such as autonomous driving and surveillance, is the focus of many models that generally fall into three frameworks: enhancement-then-detection, end-to-end approaches, and transformer-based architectures.

The enhancement-then-detection framework combines enhancement models (e.g., RetinexNet, Zero-DCE) with detection algorithms, such as YOLO. However, challenges such as feature mismatch, which refers to the discrepancy between the features learned by the enhancement model and those required by the detection algorithm, arise due to separate optimization.

End-to-end approaches, such as MPE-DETR Xue et al. (2024) and FSDNet Hu et al. (2021), integrate enhancement with detection but often suffer from static fusion techniques, which are fixed and not adaptable to different scenarios, limiting their adaptability.

Recent transformer models, which enhance low-light images using Retinex theory, show promise but face significant challenges, similar to CNNs, in processing and distinguishing illumination from reflectance. This underscores the need for further research and innovation in this area.

To address these issues, RFD-YOLO in this study introduces decomposition and fusion based on the Retinex theory within the detection process, successfully bridging the gap between low-light image restoration and object recognition.

## 3 METHODS

### 3.1 CONSIDERATION AND MOTIVATION

Traditional approaches to low-light object detection often involve a distinct phase of image enhancement performed before the actual detection. However, frequently results in a misalignment between the objectives of enhancement and the specific requirements of detection tasks. As a consequence, the overall performance can suffer in realistic, low-light conditions. To address these limitations, we draw inspiration fron Retinex theory Land & McCann (1971), which decomposes an image into reflectance and illumination component allowing object details to be separated from varying lighting conditions. This separation enables more robust detection under challenging illumination. Based on this principle, we propose an innovative end-to-end framework designed to intergrate these elements and enhance detection accuracy.

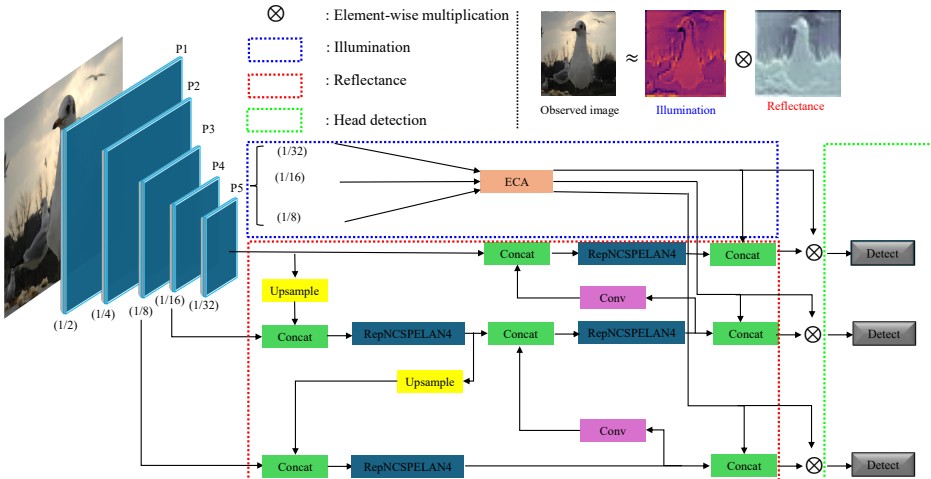

Figure 1: The architecture of the proposed RFD-YOLO (Retinex Feature Decomposition YOLO) for low-light object detection is outlined. This network incorporates a Retinex-inspired decomposition module, designed to separate illumination and reflectance components from the input image. The resulting decomposed features are then fused and fed into the detection head, thereby facilitating improved object detection capabilities in low-light environments with adversarial conditions.

The architecture of our approach, illustrated in Figure 1, encompasses two principal components that work in tandem: (1) a deep multi-scale feature aggregation module that adeptly captures hierarchical features from the levels $P_3$, $P_4$, and $P_5$, and (2) a reflectance-guided fusion pathway that synthesizes illumination-aware representations through sophisticated element-wise modulation. Together, these components aim to replicate the essence of Retinex-feature decomposition within the feature space, enabling a more nuanced understanding of the images. The following subsections provide a detailed exploration of each component, outlining the mechanisms by which they contribute to robust object detection, regardless of the presence of low-light training images.

## 3.2 DEEP MULTI-SCALE FEATURE AGGREGATION

Modern convolutional architectures generate hierarchical feature representations at various semantic levels, allowing for a detailed analysis of input images. In our method, we focus on three key feature maps referred to as $P_3$, $P_4$, and $P_5$ extracted from intermediate layers of the backbone network. These maps correspond to progressively reduced spatial resolutions and deeper receptive fields: $P_3$ identifies intricate textures, $P_4$ captures mid-level patterns, and $P_5$ conveys broader high-level contextual semantics Lin et al. (2016); Liu et al. (2018).

The choice of these feature maps is particularly relevant in low-light conditions, where local object details and overall illumination significantly impact visual perception. Specifically, $P_3$ models localised reflectance variations, while $P_5$ provides insights into the larger patterns of illumination. By integrating these different scales, our approach allows for a cohesive representation of both reflectance and illumination characteristics, which is essential for a Retinex-inspired decomposition framework.

To create a unified representation, we aggregate the features $\{P_3, P_4, P_5\}$ through both spatial alignment and channel integration. Each feature map is resized to a uniform spatial resolution and then combined using a channel-wise operation:

$$F_{\text{agg}} = \mathcal{A}(P_3, P_4, P_5), \tag{1}$$

where $\mathcal{A}(\cdot)$ denotes the aggregation function. This function may utilize techniques like upsampling, concatenation, or summation, depending on the implementation specifics.

The resulting feature $F_{\mathrm{agg}}$ provides a comprehensive, multi-scale representation of the scene, effectively integrating both detailed and broad information. This aggregated feature forms the basis for the subsequent reflectance-guided fusion stage, enhancing the model's ability to learn object representations that are invariant to variations in illumination.

### 3.3 REFLECTANCE-GUIDED FEATURE FUSION

The proposed architecture is based on the application of the Retinex model, which is designed to decompose images into their fundamental illumination $L(x, y)$ and reflectance $R(x, y)$ components. This model can be expressed as follows:

$$I(x, y) = L(x, y) \otimes R(x, y), \tag{2}$$

where $I(x, y)$ represents the observed image, $L(x, y)$ relates to variable illumination conditions and $R(x, y)$ corresponds to intrinsic reflectance—indicating stable object features Land & McCann (1971). The operation $\otimes$ indicates element-wise multiplication, demonstrating that pixel intensity at each coordinate $(x, y)$ arises from the product of the associated reflectance and illumination values.

To apply this decomposition effectively within the deep feature space, the approach utilizes two branches, each with a specific function. The illumination maps $L(x, y)$ are generated from multi-scale features $P_3, P_4$, and $P_5$ through the Efficient Channel Attention (ECA) mechanism. ECA extracts illumination features by applying a fast 1D convolution over globally pooled channels, capturing local cross-channel interactions without dimensionality reduction. This preserves essential intensity patterns while maintaining spatial structure. Such lightweight attention is crucial in low-light conditions, where accurate illumination modeling enhances object distinction in poorly lit scenes.

Conversely, the reflectance maps $R(x, y)$ are developed using a cross-scale feature enhancement pathway. This pathway integrates upsampling, concatenation, and a series of RepNCSPELAN4 blocks. Derived from the YOLOv9 framework Wang & Liao (2024), RepNCSPELAN4 distinguishes itself from conventional C2f or ELAN modules by employing reparameterised convolutions and nested connections Zhang et al. (2024). This architectural choice is significant for preserving structural details and achieving computational efficiency, both of which are crucial for accurate reflectance modelling.

The selection of RepNCSPELAN4 is based on its ability to capture hierarchical spatial patterns while minimizing aggressive downsampling. This characteristic is essential for maintaining edge integrity, object contours, and texture coherence across varying lighting conditions. The reflectance-aware features produced are further refined through lightweight convolutional layers, including ECA, which enhances spatial coherence and facilitates robust, illumination-invariant representations.

The fusion of the segregated reflectance and illumination components occurs through element-wise multiplication, consistently applied across three distinct resolution levels:

$$F_i^{\mathrm{fused}} = R_i \otimes L_i, \quad i \in \{3, 4, 5\}, \tag{3}$$

This operation reflects the core Retinex principle within the deep feature space, enabling the network to retain high-confidence structural features while minimizing noise and distortions from inconsistent lighting.

This reflectance-guided fusion methodology represents a significant advancement in the field, offering various benefits. It enables the model to focus on stable, intrinsic object characteristics, regardless of fluctuating lighting, thereby enhancing generalization in low-light scenarios. Additionally, it introduces an efficient and fully differentiable mechanism that integrates illumination awareness into the detection process. This framework eliminates the need for external enhancement modules or reliance on paired low-light datasets, addressing limitations commonly found in traditional enhancement-plus-detector pipelines by fostering genuinely illumination-invariant features within a unified, end-to-end learning paradigm.

# 4 EXPERIMENT RESULTS

To demonstrate the robustness of our approach, we evaluate the proposed method in adverse visual contexts, focusing on foggy and low-light conditions.

## 4.1 DATASETS

In our model training endeavours, we utilize the comprehensive training and validation splits derived from both the Pascal VOC 2007 and 2012 datasets Everingham & Winn (2009); Everingham et al. (2012), which encompass 20 distinct classes. The test split from VOC 2007 comprises a substantial collection of 4,952 images, which we meticulously adapt to align with the class nomenclature of two additional datasets: ExDark, featuring 10 classes, and RTTS, consisting of 5 classes Loh & Chan (2019) Li et al. (2017). This alignment process involves a careful category mapping, ensuring that only overlapping classes are retained, while unmatched categories are excluded from our evaluation within the VOC_Norm training set.

We designate the revised VOC 2007 test set, following this class mapping, as the VOC_N_Ts (VOC Normal Test Set). To rigorously evaluate the model's resilience across a spectrum of challenging conditions, we create deliberately complex test images and apply two specialised filters to the original VOC 2007 test set:

- **VOC_F_Ts (Foggy Filter Test Set)**: To replicate the effects of foggy conditions, we employ the atmospheric scattering model (ASM) Narasimhan & Nayar (2002), which artfully introduces a layer of white haze to each image. The parameters dictating the intensity and spread of the fog are randomly selected, yielding a diverse array of fog densities that range from subtle mist to dense, obscuring fog.

- **VOC_D_Ts (VOC Dark Test Set)**: For simulating low-light environments, we craft synthetic dark images by elevating each normalised clear image to a randomly chosen gamma value between 1.5 and 5. This transformation, defined by the equation $I_{dark} = I^{\gamma}$, results in varying degrees of darkening, creating a rich spectrum of visibility challenges based on the sampled gamma values.

These innovative filters enable us to generate additional foggy and low-light variants of the VOC test images, which are then used for evaluation alongside real-world datasets, ExDark and RTTS. For our primary assessment, we treat ExDark and RTTS as independent test sets, ensuring that our model does not gain prior access to these datasets during the training phase. Furthermore, the class labels within ExDark and RTTS are meticulously mapped to correspond with the Pascal VOC categories, thereby maintaining a consistent framework across all evaluation metrics.

## 4.2 IMPLEMENTATION DETAILS

In this study, all images were uniformly resized to a resolution of $640 \times 640$ pixels, ensuring consistency across both the training and testing phases. The models were trained using a carefully selected batch size of 2, spanning a total of 80 epochs. To enhance the learning process, the training commenced with an initial learning rate of 0.01, which was systematically reduced to 0.0001, allowing for finer adjustments as the training progressed. The optimization employed was stochastic gradient descent (SGD), complemented by a weight decay of $5 \times 10^{-4}$ to effectively mitigate the risk of overfitting and improve generalization on unseen data.

# 5 RESULTS AND ANALYSES

## 5.1 QUALITATIVE ANALYSIS

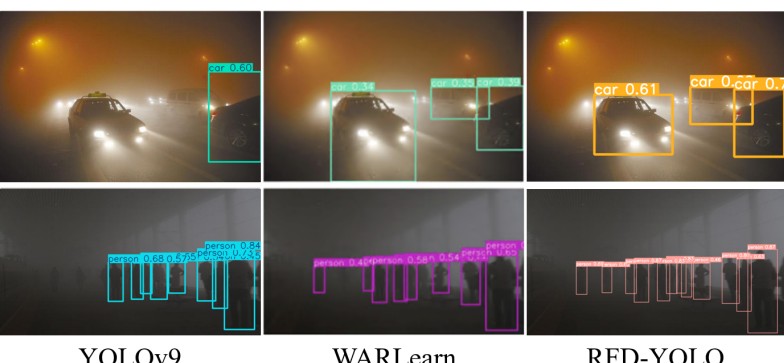

| YOLOv9 | WARLearn | RFD-YOLO |

Figure 2: Detection results on fog-affected RTTS dataset using YOLOv9, the current state-of-the-art WARLearn, and the proposed RFD-YOLO. RFD-YOLO achieves superior detection accuracy and confidence under low-visibility conditions. This evaluation is best appreciated at 4× magnification.

In our assessment of the detection performance of the RFD-YOLO model, we conducted a comparative analysis against both the standard YOLOv9 and the more sophisticated WARLearn framework Agarwal et al. (2025), with an emphasis on challenging conditions such as low-light environments and scenarios impacted by fog. The detection results for each framework are illustrated in Figures 2 and 3, highlighting their performance in visually analogous contexts. Additional ablation experiments on our proposal and individual modules are included for completeness. Please refer to the Appendix for full results and discussion A.

Our findings reveal that RFD-YOLO consistently outperforms both YOLOv9 and WARLearn in terms of localization accuracy and robust detection capabilities, particularly for small, occluded, or low-contrast objects. Compared to these alternatives, RFD-YOLO generally demonstrates a lower false positive rate and provides more comprehensive detection results. These qualitative insights underscore the enhanced generalization and reliability of RFD-YOLO in adverse weather conditions and variable lighting environments. A detailed quantitative evaluation of the performance enhancements provided by RFD-YOLO will follow in the subsequent analysis.

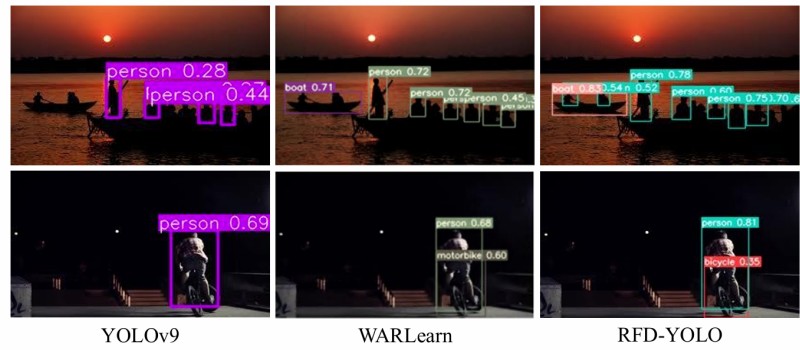

| YOLOv9 | WARLearn | RFD-YOLO |

Figure 3: Visual comparison of detection performance on low-light ExDark dataset. From left to right: baseline YOLOv9, the current state-of-the-art WARLearn, and the proposed RFD-YOLO. RFD-YOLO demonstrates higher accuracy and confidence, particularly under challenging illumination conditions. The results are best appreciated at a 4× magnification.

## 5.2 QUANTITATIVE ANALYSIS

This section offers a quantitative evaluation of RFD-YOLO's performance, benchmarking it against leading models on complex foggy and low-light datasets using the Mean Average Precision (mAP) metric.

Table 1: A comparison of our method with previously established approaches is carried out on fog-related benchmarks. The mean Average Precision (mAP) scores for the datasets V_N_Ts (VOC_Normal_test), V_F_Ts (VOC_Foggy_test), and RTTS are presented in the final three columns. This analysis provides valuable insights into the robustness of each model under varying visibility conditions.

| Method | Train data | Voc_N_Ts | Voc_F_Ts | RTTS |
|---|---|---|---|---|
| Yolov12 | Hybrid | 66.1 | 66.0 | 31.30 |
| MSBDN | VOC_Norm | - | 57.38 | 30.20 |
| GridDehaze | VOC_Norm | - | 58.23 | 31.42 |
| DAYolo | Hybrid | 56.51 | 55.11 | 29.93 |
| DSNET | Hybrid | 53.29 | 67.40 | 28.91 |
| IA-Yolo | Hybrid | 73.23 | 72.03 | 37.08 |
| GDIP-Yolo | Hybrid | 73.70 | 71.92 | 42.42 |
| DETR | Hybrid | 73.7 | 72.7 | 41.9 |
| WARLearn | Hybrid | 69.11 | 75.10 | 52.60 |
| **RFD-YOLO (Our)** | VOC_Norm | **83.2** | **78.4** | 52.50 |

**Foggy conditions:** Table 1 presents a comparative analysis of RFD-YOLO's detection capabilities against state-of-the-art methods across fog-specific benchmarks, including the VOC regular test set (Voc_N_Ts), the VOC foggy test set (Voc_F_Ts), and the RTTS. The term 'Hybrid' denotes models trained on a mix of clear and foggy images. Our approach, however, leverages the VOC_Norm dataset exclusively, enhancing detection robustness under adverse conditions. RFD-YOLO achieved the highest mAP across all benchmarks: 83.2 on Voc_N_Ts, 78.4 on Voc_F_Ts, and 52.5 on RTTS, surpassing recent techniques such as WARLearn, DETR Zhao et al. (2024), DA-YOLO Zhang et al. (2021a), and IA-YOLO Liu et al. (2021). This superior performance is attributed to our Retinex-inspired feature extraction, which effectively yields reliable results in challenging environmental conditions, even when trained solely on normal images.

Table 2: Evaluation of current methodologies against established dark-related benchmarks is presented. The mean Average Precision (mAP) scores for VOC_N_Ts (VOC_Normal_test), V_D_Ts (VOC_Dark_test), and Exdark are summarised in the final three columns, offering a detailed analysis of each model's performance and resilience across different visibility scenarios.

| Method | Train data | Voc_N_Ts | Voc_D_Ts | Exdark |
|---|---|---|---|---|
| Yolov12 | Hybrid | 57.0 | 51.9 | 40.2 |
| ZeroDCE | VOC_Norm | - | 33.57 | 37.03 |
| DAYolo | Hybrid | 41.68 | 21.53 | 18.15 |
| DSNET | Hybrid | 64.08 | 43.75 | 36.97 |
| IA-Yolo | Hybrid | 56.01 | 48.44 | 26.67 |
| GDIP-Yolo | Hybrid | 63.23 | 57.85 | 42.56 |
| DETR | Hybrid | 59.2 | 57.2 | 48.3 |
| WARLearn | Hybrid | 75.50 | 70.90 | 55.70 |
| **RFD-YOLO (Our)** | VOC_Norm | 73.6 | 69.6 | **58.8** |

**Low-light conditions:** As shown in Table 2, RFD-YOLO reaches a leading mAP of 58.8 on the ExDark dataset, outperforming competitors like WARLearn (55.7) and DETR (48.3). Notably, RFD-YOLO achieves this without low-light images in its training regimen, underscoring the efficacy of our Retinex-based methodology in developing illumination-invariant features. This attribute enables strong generalization in real-world low-light situations, affirming RFD-YOLO as a robust solution for object detection under challenging lighting conditions.

**Impact Analysis Across Low-Light and Fog Intensities:** While RFD-YOLO demonstrates strong resilience in low-light and foggy environments, significant challenges arise when visibility plummets below critical thresholds. In scenarios characterised by extremely dense fog or total darkness, the reliability of detection diminishes, not due to inherent model deficiencies, but rather the scarcity of

visual information. This limitation is a common struggle for all vision-based methods. Pixel values tend to consolidate at brightness extremes, diminishing contrast and obscuring object features, complicating even human recognition. RFD-YOLO maintains high accuracy up to these thresholds, surpassing competing models; however, performance declines universally beyond this point. These results reveal fundamental limitations in object detection capabilities under severely degraded conditions.

## 5.3 REAL-TIME PERFORMANCE

In our evaluation of real-time performance, as illustrated in Table 3, RFD-YOLO demonstrates an exceptional throughput of 88.82 frames per second (FPS) when deployed on a single NVIDIA RTX 4090 GPU. This remarkable performance not only sets a new standard but also surpasses several prominent benchmarks in the field. Notably, it outshines YOLOv12, which operates at 84.12 FPS, and DETR, which achieves a mere 49.72 FPS—both of which leverage transformer architectures for object detection.

Table 3: Comparison of FPS performance between the proposed model and existing detection methods. RFD-YOLO achieves the highest inference speed.

| Model | Yolov12 | IA-YOLO | DETR | GDIP-YOLO | DA-YOLO | RFD-YOLO |
|-------|---------|---------|-------|-----------|---------|----------|
| FPS | 84.12 | 22.84 | 49.72 | 29.78 | 24.27 | 88.82 |

Moreover, RFD-YOLO's efficiency surpasses that of DA-YOLO, renowned for its prowess in foggy object detection, achieving a throughput of 24.27 FPS, and IA-YOLO, a leading contender for low-light scenarios, which runs at 22.84 FPS. These comparisons reveal that RFD-YOLO not only excels in maintaining high detection accuracy but also offers unparalleled real-time performance capabilities, making it a formidable choice for real-time object detection tasks across various challenging environments.

## 6 CONCLUSION

In this study, we introduced RFD-YOLO, an innovative object detection framework designed to address the challenges presented by fog and low-light environments. The framework features a Retinex-inspired illumination module and an element-wise operation that simulates the interaction between illumination and surface reflectance. This design approach enhances feature representation, making it more robust and adaptable. Extensive evaluations on benchmark datasets under foggy and low-light conditions demonstrate that RFD-YOLO consistently outperforms leading state-of-the-art methods, even when trained exclusively on images captured under standard lighting conditions.

Ablation studies highlight the significance of explicit illumination modelling and Retinex-based feature interactions, which contribute to the model's adaptability to unfamiliar adverse conditions. Additionally, our findings highlight an important aspect: the variety in training and evaluation datasets significantly impacts the model's generalization capability and detection accuracy. While our framework exhibits notable resilience against environmental degradation, extreme visibility loss or poor illumination can still lead to undetectable objects, a challenge that humans also face in such scenarios.

This observation highlights the need for ongoing research to develop more sophisticated architectures and thoroughly characterise the diverse range of real-world visual variations. Looking ahead, we plan to enhance our methodology to manage even more complex atmospheric conditions and lighting scenarios. We also aim to broaden the scope of our training images to include various scene types and quality levels, thereby improving the model's robustness. By diversifying the training dataset with a richer assortment of examples, we aim to enhance the model's ability to identify objects across various real-world situations, regardless of environmental changes.

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

## A  APPENDIX

### A.1  ABLATION STUDY

This section presents a comprehensive ablation study that analyses the individual and collective effects of key components within the architecture. The focus is on clarifying how each module—specifically the Retinex-inspired illumination module and the element-wise operation—contributes to the model's performance in challenging visual scenarios. All ablation experiments utilize a combination of standard and hybrid datasets, incorporating both clear and adverse

Table 4: Ablation Study with Exdark dataset.

| Method | Train data | Voc_N_Ts | Voc_D_Ts | Exdark |
|---|---|---|---|---|
| Without illumination | **Hybrid** | 51.6 | **60.8** | **52.4** |
| | VOC_Norm | 62.3 | 52.7 | 41.4 |
| Without element-wise | **Hybrid** | **72.3** | **68.6** | **57.5** |
| | VOC_Norm | 69.6 | 66.3 | 55.6 |
| **RFD-YOLO** | **Hybrid** | **74.0** | **70.2** | **60.0** |
| | VOC_Norm | 73.6 | 69.6 | 58.8 |

conditions (e.g., foggy or low-light images), to ensure fair comparisons with existing approaches and accurately evaluate each component's contribution to generalization.

**Influence of the Illumination Module:**

The illumination module, based on Retinex theory, is implemented to model lighting variations, allowing the network to separate illumination from reflectance. This separation is crucial for extracting illumination-invariant features necessary for detection in challenging environments. To assess its contribution, the illumination module was removed from the architecture, and the model was retrained using the same hybrid data configuration. Results in Table 4 and 5 indicate a significant drop in performance upon omitting the illumination module; for example, the mean Average Precision (mAP) on the RTTS dataset decreased from 52.5 to 35.5 and from 58.8 to 41.4 on ExDark. This demonstrates the importance of explicit illumination modeling in adapting to low-light and foggy conditions, even when challenging imagery is present during training. The hybrid training data also appears to influence these outcomes, as reflected in the highlighted results.

Table 5: Ablation Study with RTTS dataset.

| Method | Train data | Voc_N_Ts | Voc_F_Ts | RTTS |
|---|---|---|---|---|
| Without illumination | **Hybrid** | **75.4** | **74.8** | **42.4** |
| | VOC_Norm | 74.0 | 58.6 | 35.5 |
| Without element-wise | **Hybrid** | **83.6** | **77.6** | **52.5** |
| | VOC_Norm | 79.7 | 79.3 | 48.3 |
| **RFD-YOLO** | **Hybrid** | **85.1** | **80.5** | **52.9** |
| | VOC_Norm | 83.2 | 78.4 | 52.5 |

**Contribution of Element-wise Operation with Diversity Image:** The element-wise operation in this design aligns with the Retinex principle, considering each image as a product of illumination and reflectance (image $\approx$ illumination $\otimes$ reflectance). This operation enables the network to estimate and integrate these components independently during feature extraction, facilitating illumination-aware representations. When this feature is excluded, the network's ability to adjust for lighting variations at the feature level is compromised, reverting to a conventional architecture without illumination enhancement capabilities. As shown in Table 4 and 5, this results in further declines in mAP for both RTTS and ExDark datasets, indicating the significance of the Retinex-inspired operation for managing diverse illumination and maintaining detection performance in complex environments.

