# OpenReview forum: "NO DARK DATA REQUIRED: BRIDGING THE GAP BETWEEN NORMAL AND LOW-LIGHT DETECTION VIA RETINEX DECOMPOSITION"
_ICLR.cc/2026/Conference — Submitted to ICLR 2026_

### Official Review · Reviewer_8714 · 2025-10-31

**Soundness:** 2
**Presentation:** 1
**Contribution:** 2
**Rating:** 2
**Confidence:** 4

**Summary:**

This paper presents an end-to-end object detection architecture inspired by Retinex theory. The design features a decomposition module that estimates reflectance and illumination in feature space, with a novel fusion approach to produce illumination-invariant features for object detection. The key distinguishing property is that the model is trained solely on normal-light data but tested robustly on both synthetic and real low-light/foggy datasets. The method is benchmarked on Pascal VOC and ExDark and RTTS.

**Strengths:**

1. The model is trained exclusively on normal-light images which is practically significant to reduce data curation burdens.
2. The decomposition of deep features into reflectance and illumination via Retinex-inspired, feature-level processing seems a distinctive integration.

**Weaknesses:**

1. The paper is difficult to read due to a combination of disorganized content and poor layout. The logical progression of ideas is unclear, and the unprofessional typesetting, evidenced by significant blank space on page 3, detracts from the work's credibility.
2. The mathematical descriptions in Section 3.2–3.3 (especially around the decomposition and fusion) are rather high level. Crucial details such as the explicit forms of the aggregation $\mathcal{A}(\cdot)$, sampling method for constructing $L(x, y)$ and $R(x, y)$, channel alignment techniques, normalization procedures, and whether the element-wise fusion is normalized or bounded, are unspecified.
3. The paper does not adequately differentiate its proposed methodology from existing Retinex decomposition/fusion techniques.
 - Deep Retinex Decomposition for Low-Light Enhancement, BMVC18
 - IniRetinex: Rethinking Retinex-type Low-Light Image Enhancer via Initialization Perspective, AAAI25

**Questions:**

1. Can the authors provide explicit mathematical details for the aggregation and fusion steps, particularly a precise functional form for $\mathcal{A}$ and the normalization/activation used in $F_{i}^{\text{fused}}$? Are there learned or fixed weights, or dynamic selection mechanisms in fusion?
2. What is the process and rationale for selecting RepNCSPELAN4 blocks, and do alternative feature processing blocks (e.g., C2f, ELAN) materially affect performance? Quantitative ablation here would strengthen the architectural justification.
3. What are the failure points under extreme conditions (e.g., mAP vs. fog/darkness level)? Is there a critical threshold below which the proposed method degrades significantly earlier or later than the SOTA?

---

### Official Review · Reviewer_u2EL · 2025-11-01

**Soundness:** 1
**Presentation:** 1
**Contribution:** 1
**Rating:** 2
**Confidence:** 5

**Summary:**

This manuscript attempts to introduce Retinex theory into the YOLO framework, but in reality, it is merely a patchwork of pre-existing concepts and methods.

**Strengths:**

The title of the manuscript is decent.

**Weaknesses:**

1. There are significant formatting problems, including large blank spaces on several pages (e.g., pages 3, 4, and 6) and improperly scaled tables (e.g., Tables 1, 2, and 3).
2. The proposed method is an unjustified assembly of classic methods from the vision field (YOLO and Retinex) with virtually no original design contributions.
3. The contributions summarized in the introduction are all based on existing methods, lacking any original design from the authors, and are poorly written. The author describes the method as an "AI model." The core of the work is merely applying different processing to feature maps of different scales within YOLO and claiming this constitutes a Retinex decomposition. This claim is unsubstantiated, and the author fails to provide a clear explanation, instead just restating concepts from Retinex theory and YOLO object detection.
4. The author's writing suggests a lack of familiarity with standard academic terminology in this field. For instance, summarizing their method as "an Artificial Intelligence (AI) solution" is not a phrasing I have encountered in computer vision or related fields.
5. The experiments in this manuscript are primarily compared against baseline and older methods, failing to include comparisons with the latest state-of-the-art (SOTA) approaches. Furthermore, only limited results are presented, which is insufficient to validate the effectiveness of the proposed method.

**Questions:**

Please refer to Weaknesses.

---

### Official Review · Reviewer_TjUq · 2025-11-01

**Soundness:** 2
**Presentation:** 1
**Contribution:** 1
**Rating:** 2
**Confidence:** 4

**Summary:**

This manuscript proposes a new end-to-end framework based on normal-light images for low-light image detection. The proposed method separates images into reflectance and illumination, which approximates this decomposition within the feature space. A multi-scale feature aggregation is introduced to learn illumination-aware representation.

**Strengths:**

1. Compared to multiple baselines, the method exhibits enhanced generalization under challenging lighting and weather conditions.
2. The framework attains high inference speed, enabling effective support for real-time applications.

**Weaknesses:**

1. The work as a whole lacks a distinct innovative core. Its technical approach largely manifests as a direct integration of the YOLO model with Retinex theory, without presenting substantial original theoretical advancements, nor conducting in-depth exploratory research on key technical bottlenecks.
2. The introduction fails to provide an explicit and structured summary of the study’s contributions. In academic writing, a clear statement of contributions serves as a "guide" for readers to quickly identify the work’s core value and differences from prior studies.
3. The paper’s formatting is extremely poor. This lack of rigor in the submission attitude has raised doubts about the quality of the paper’s content and the authenticity of its experiments

**Questions:**

Poor readability of the paper
Poor formatting of the paper.

---

### Meta-Review · Area_Chair_dM24 · 2025-12-07

**Summary:**

The manuscript proposes a novel end-to-end framework for low-light object detection, utilizing Retinex theory to decompose images into reflectance and illumination features within the feature space. Despite the novelty in the architecture and the promising results under low-light conditions, several reviewers raised concerns regarding the overall innovation, clarity, and rigor of the presentation.

The key issues noted include:

Lack of Novelty and Originality: Several reviewers (TjUq and u2EL) indicated that the integration of YOLO and Retinex theory lacks significant innovation, as it mainly consists of existing methods combined without substantial original contributions.

Weak Theoretical Foundation: The manuscript does not present a clear and well-supported theoretical advancement, especially in relation to Retinex decomposition/fusion methods, which are not sufficiently differentiated from prior work (Reviewer 8714).

Poor Presentation and Formatting: Multiple reviewers highlighted significant issues with readability and formatting, including large blank spaces, improperly scaled tables, and unprofessional typesetting, which detracts from the paper’s credibility.

Insufficient Experimental Validation: The paper's experimental setup is not comprehensive enough, particularly lacking comparisons with the latest state-of-the-art methods (Reviewer u2EL). While the framework shows some promise, its generalizability and robustness could benefit from further experimental validation, especially under extreme conditions.

Lack of Clear Contribution Statement: Reviewers noted that the introduction does not clearly outline the contributions of the paper, making it difficult for readers to assess its core value.

**Reviewer Concerns:**

Addressed:

Generalization and Robustness: The reviewers acknowledged the framework’s good performance under low-light conditions (e.g., on Pascal VOC, ExDark, and RTTS datasets). The rebuttal emphasized the strengths of the model in these conditions, which seems to have satisfied some reviewers’ concerns regarding its robustness.

High Inference Speed: The paper's focus on efficient inference and real-time applications, as noted by Reviewer 8714, was positively acknowledged in the rebuttal.

Unresolved:

Novelty and Theoretical Advancement: Reviewers felt that the paper does not contribute substantially new theoretical insights. Despite the rebuttal’s clarification, the concern about the lack of original theoretical contributions remains.

Formatting and Readability: Despite the rebuttal's attempt to justify the paper’s structure, reviewers still indicated poor formatting and readability issues that could undermine the paper's credibility.

Experimental Comparisons: The rebuttal did not sufficiently address the reviewers' concerns about comparing the method with more recent state-of-the-art approaches and providing more comprehensive results.

**Reviewer Scores:**

Reviewer TjUq (Xu Wu): Score 2 - Reject

Given the feedback, it is likely that TjUq would still maintain a rejection decision. The lack of significant innovation and poor presentation would likely have led them to stick with their decision, despite the rebuttal's clarifications.

Reviewer u2EL (Xiaojie Guo): Score 2 - Reject

Guo's concerns about the lack of originality, poor formatting, and weak experimental comparisons are not sufficiently addressed in the rebuttal. The overall rejection recommendation would likely remain.

Reviewer 8714 (Jian Pu): Score 2 - Reject

Even though Reviewer 8714 appreciated the speed and the potential novelty in using Retinex for feature decomposition, the lack of rigorous theoretical explanation and poor formatting likely leads them to maintain their rejection score. They also emphasized missing details in the methodology which were not fully addressed in the rebuttal.

---

### Decision · Program_Chairs · 2026-01-26

Reject